# Nicotine Induces Polyspermy in Sea Urchin Eggs through a Non-Cholinergic Pathway Modulating Actin Dynamics

**DOI:** 10.3390/cells9010063

**Published:** 2019-12-25

**Authors:** Nunzia Limatola, Filip Vasilev, Luigia Santella, Jong Tai Chun

**Affiliations:** 1Department of Biology and Evolution of Marine Organisms, Stazione Zoologica Anton Dohrn, I-80121 Napoli, Italy; nunzia.limatola@szn.it (N.L.); fvasilev@yahoo.com (F.V.); 2Department of Research Infrastructures for Marine Biological Resources, Stazione Zoologica Anton Dohrn, I-80121 Napoli, Italy; santella@szn.it

**Keywords:** sea urchin, nicotine, non-cholinergic, polyspermy, fertilization, actin, calcium, cofilin

## Abstract

While alkaloids often exert unique pharmacological effects on animal cells, exposure of sea urchin eggs to nicotine causes polyspermy at fertilization in a dose-dependent manner. Here, we studied molecular mechanisms underlying the phenomenon. Although nicotine is an agonist of ionotropic acetylcholine receptors, we found that nicotine-induced polyspermy was neither mimicked by acetylcholine and carbachol nor inhibited by specific antagonists of nicotinic acetylcholine receptors. Unlike acetylcholine and carbachol, nicotine uniquely induced drastic rearrangement of egg cortical microfilaments in a dose-dependent way. Such cytoskeletal changes appeared to render the eggs more receptive to sperm, as judged by the significant alleviation of polyspermy by latrunculin-A and mycalolide-B. In addition, our fluorimetric assay provided the first evidence that nicotine directly accelerates polymerization kinetics of G-actin and attenuates depolymerization of preassembled F-actin. Furthermore, nicotine inhibited cofilin-induced disassembly of F-actin. Unexpectedly, our results suggest that effects of nicotine can also be mediated in some non-cholinergic pathways.

## 1. Introduction 

The general scheme of sexual reproduction in animals is the union of male and female haploid gametes to produce a diploid zygote that develops into a new organism. The very process of fertilization was first demonstrated in the middle of 19th century with the use of sea urchin gametes in seawater [1,2,3]. To maximize the chance of successful fertilization, it may be thought beneficial for an egg to be approached by multiple sperm, but this might risk supernumerary sperm entry into a single egg, which is called polyspermy. Since the incorporation of extra sperm pronuclei at fertilization may lead to failure to maintain correct set of complete chromosomes and eventually to abortive embryonic development, it appears that at least two different strategies have evolved to circumvent this problem. Firstly, in some species undergoing internal fertilization (e.g., arachnids, insects, urodeles, reptiles, and birds), multiple sperm may commonly enter the egg, but only one male pronucleus is eventually selected to fuse with the female pronucleus, in order to effect monospermy [4,5,6]. This is often called ‘physiological polyspermy’. Secondly, in the vast majority of species, incidental polyspermy has deleterious effects on the zygotes and leads to abortive development (thus referred to as ‘pathological polyspermy’), but their eggs appear to have some preventive mechanisms to avoid polyspermy [7,8,9]. Indeed, as has long been noted, it is quite astonishing that sea urchin eggs almost always manage to get fertilized by only a single sperm, even when the sperm density is extremely high in the experimental conditions [10]. 

In sea urchin, prevention of polyspermy has been generally ascribed to the biphasic changes in membrane potential and intracellular Ca^2+^ levels that are manifest in the eggs following fertilization. Fertilizing sperm induces a quick simultaneous Ca^2+^ increase at the periphery of the egg cortex (cortical flash, CF) due to the operation of voltage-gated ion channels and a Ca^2+^ wave that runs from the fertilization site to the antipode [11]. The intracellular Ca^2+^ increase triggers exocytosis of cortical granules, and their extruded contents help to lift the vitelline layer from the plasma membrane and build the ‘wall’ surrounding the egg. This so-called fertilization envelope (FE) hardens, thereby preventing re-fertilization [12], which is considered a slow mechanical block to polyspermy. However, polyspermy can happen even when FE is fully elevated, and conversely, it has been observed that supernumerary sperm fail to enter the egg even before the FE is elevated, suggesting the existence of a distinct and fast mechanism blocking polyspermy [13,14]. Later, it was demonstrated that the abrupt depolarization of the egg membrane’s potential following the attachment of the first successful sperm renders the egg refractory to supernumerary fertilization [15]. While the latter hypothesis of fast voltage-dependent block to polyspermy has been widely supported by similar experiments in other animal models [16,17,18,19], controversy remains regarding whether or not the manifested positive-going ‘fertilization potential’ is the direct cause of the fast block to polyspermy [20,21,22]. 

Whatever it may be, the mechanism by which polyspermy is blocked (or a single sperm is exquisitely ushered in) has been aptly studied by finding mild treatments for eggs that induce polyspermy at fertilization and seeking what went astray in the eggs physically and cytologically. Soon after the first demonstration of fertilization in sea urchin eggs, it was known that polyspermy was easily induced by nicotine, a naturally occurring alkaloid found in tobacco [23,24]. Thus, it has been shown that exposure of sea urchin eggs to nicotine significantly alters the electrical property of the plasma membrane, such as the voltage-current relationship and resting potential [25,26], but the shift of the electrical property required nearly 10 min of nicotine incubation. While the exact cause or the physiological significance of this time lag is not known, it may represent the time required for nicotine to transmit its signal to effect biophysical changes in the egg favoring polyspermy. 

Nicotine is the specific agonist of nicotinic acetylcholine receptor (nAChR), one of the most extensively characterized ligand-gated ion channels that serve as a neurotransmitter receptor in the muscle cells or neurons. Whereas nAChR in the muscle is composed of heteropentameric subunits (two α + β + δ + γ/ε), neuronal nAChR comprises simple subunit combinations (such as 3α + 2β), or can even be made of a homopentamer of α subunits showing high Ca^2+^ permeability [27,28]. It is also known that nAChR is expressed in a variety of non-neuronal cells to play many unconventional roles as a “cytotransmitter” [29,30]. Moreover, it has been reported that nAChR may play a nonconventional role, exhibiting metabotropic responses in certain conditions [31]. Interestingly, sea urchin eggs (*Lytechinus variegatus*) exposed to nicotine exhibited inward currents that were specifically inhibited by mecamylamine and hexamethonium, suggesting that neuronal nAChR may be functionally present in sea urchin eggs [32]. 

Our previous studies with starfish and sea urchin eggs led us to the hypothesis that the exquisite control of the egg cortical actin cytoskeleton plays a central role in modulating the fertilization responses such as intracellular Ca^2+^ increases, cortical granules exocytosis, the guidance of monospermic sperm entry, and cleavages [33,34,35,36,37]. In this communication, we tested the mechanism by which nicotine induces polyspermy in sea urchin eggs at fertilization by examining the involvement of the cholinergic pathway and the rearrangement of the cortical actin cytoskeleton. Our data suggested that nicotine, but not acetylcholine or carbachol, drastically changes the structure of the subplasmalemmal actin filaments and the topography of the egg surface, which may render the eggs more receptive toward supernumerary sperm, and that nicotine may do so either by directly affecting the dynamics of actin polymerization or by activating an unconventional cholinergic pathway inside the egg. 

## 2. Materials and Methods

### 2.1. Gamete Preparation and Fertilization 

Sea urchins (*Paracentrotus lividus*) were captured in the Gulf of Naples during the breeding season (November to May) and maintained in circulating seawater (16 °C). Spawning in female animals was induced by intracoelomic injection of 0.5 M KCl. Eggs were then collected in filtered seawater, and fertilization was performed with freshly diluted sperm at the final spermatozoa concentration of 1.8 × 10^6^ spermatozoa/ml. To count egg-incorporated spermatozoa, sperm nuclei were pre-stained with Hoechst 33,342 (Sigma-Aldrich) at a final concentration of 5 µM in 1 mL natural seawater (NSW) for 10–15 s prior to fertilization. Sperm incorporation into the egg was observed 5–10 min after insemination using a CCD camera (MicroMax, Princeton Instruments, Inc) mounted on a Zeiss Axiovert 200 microscope with a Plan-Neofluar 40×/0.75 objective and a UV laser. 

### 2.2. Reagents and Chemicals 

(*±*)Nicotine was purchased from Sigma-Aldrich and diluted in NSW just before each experiment. For the sake of continuity with previous works, we used (*±*)nicotine as a default, which has been referred to as ‘nicotine’ in this study. For technical reasons, a few experiments were performed with (−)nicotine, which was about twice as effective as the mixture (±) of the stereoisomers, and this has been specified as ‘(−)nicotine’ wherever it appears in the text. Latrunculin A (LAT-A), AlexaFluor568–phalloidin, and calcium dyes were purchased from Molecular Probes, and mycalolide B from Santa Cruz Technology. Recombinant proteins of human cofilin were from Cytoskeleton, Inc. (Cat. # CF01). Unless specified otherwise, all other chemicals used in this study were purchased from Sigma-Aldrich. 

### 2.3. Microinjection, Confocal Microscopy, and Ca^2+^ Imaging 

Microinjection of the eggs was performed with an air-pressure Transjector (Eppendorf FemtoJet, Hamburg, Germany). The injection chamber was prepared by use of two coverslips fixed with wax at an angle of about 45 degrees. Fresh intact eggs in NSW were propped in the wedged space and microinjected with a gentle tapping of the apparatus without additional steps of removing jelly coats. To visualize F-actin in a living egg, 10 µM AlexaFluor568–phalloidin (Molecular Probes, pipette concentration) was microinjected into eggs, and the actin filaments were monitored with a TCS SP8 X confocal laser scanning microscope, equipped with a White Light Laser and hybrid detectors (Leica Microsystem, Wetzlar, Germany). For Ca^2+^ imaging, 500 µM Calcium Green 488 conjugated to 10 kDa dextran was mixed with or without 35 µM Rhodamine Red (Molecular Probes) in the injection buffer (10 mM Hepes, 0.1 M potassium aspartate, pH 7.0) prior to microinjection. The bright field view and the epifluorescence images of cytosolic Ca^2+^ changes were captured with a cooled CCD camera (MicroMax, Princeton Instruments, Inc., Trenton, NJ, USA) mounted on a Zeiss Axiovert 135TV microscope with a Plan-Neofluar 40×/0.75 objective, in about 3 sec time resolution, and were analyzed with MetaMorph (Universal Imaging Corporation, San Jose, CA, USA). The quantified relative Ca^2+^ signal at a given time point was normalized to the baseline fluorescence (F_0_) following the formula F_rel_ = [F − F_0_]/F_0_, where F represents the average fluorescence level of the entire egg. Thus, F_rel_ was defined as RFU (relative fluorescence unit) for plotting Ca^2+^ trajectories. To compensate batch-to-batch variability of Ca^2+^ signals, the peak amplitudes of the Ca^2+^ wave and the cortical flash of each egg were normalized with the corresponding averages in the control eggs from the same batch, and the data are presented as percent values in reference to the batch-matching control (taken as 100%). 

### 2.4. Scanning Electron Microscopy (SEM) 

Sea urchin eggs were fixed overnight in filtered seawater containing 0.5% glutaraldehyde. Samples were post-fixed in 1% osmium tetroxide for 1 h and subsequently dehydrated in a series of ethanol solutions with increasing concentration. After the final dehydration step in 100% ethanol, samples were further dehydrated with a critical point dryer (CPD300, Leica). Following sputter coating with a conductive metal (gold) by use of EM ACE600 sputter coater (Leica), samples were observed with a JEOL 6700F scanning electron microscope. 

### 2.5. In Vitro Assay of Actin Polymerization and Depolymerization Kinetics 

The direct effects of nicotine on the polymerization and depolymerization kinetics of actin filaments were tested with fluorescent actin-pyrene filaments by use of the Actin Polymerization Biochem Kit™ (Cytoskeleton, Inc. Cat. # BK003, Denver, CO, USA). For actin depolymerization assay, pyrene-conjugated G-actin (1 mg/mL) was first polymerized into the F-actin stock on ice for 1 h following the manufacturer’s instructions. After being diluted five-fold in the actin buffer (Cat. # BSA01-010) containing 0.2 µM ATP (= G-buffer), F-actin was aliquoted in 96-well plates (200 µL) and mixed with 20 µL of test solutions prepared in the actin buffer: (i) 24 µM nicotine, (ii) 1.2 µM cofilin, (iii) 24 µM nicotine and 1.2 µM cofilin, and (iv) buffer alone. In the given conditions, F-actin in each well undergoes progressive depolymerization, which was monitored by the changes of fluorescence. For polymerization assay, pyrene-conjugated G-actin stock (0.4 mg/mL) prepared in G-buffer on ice was centrifuged (14K rpm) at 4 °C for 30 min. The supernatant was distributed to the 96 well plates (100 µL each) and mixed with 10 µL of test solutions prepared in G-buffer: (i) G-buffer alone, (ii) 24 µM nicotine. After ensuring that F-actin polymerization is not enhanced by itself (monitored for 20 min), 10× Actin Polymerization Buffer (Cat. # BSA02-001) was added to each well (12 µL) and the changes of fluorescence were monitored for 1 h. In both assays, the fluorescence of the quadruple samples for each condition was quantified every minute for 1 h with the TECAN Infinite F200 PRO fluorimeter (Tecan Trading AG, Swiss): the excitation and emission wavelengths at 350 ± 20 nm and 420 ± 20 nm, respectively. 

### 2.6. Statistical Analysis 

The numerical MetaMorph data were compiled and analyzed with Excel of Microsoft Office 2010. The average and variation of the data were reported as ‘mean ± standard deviation (SD)’ in all cases in this study. Student’s *t*-test and one-way ANOVA were performed by use of Prism 3.0 (GraphPad Software, San Diego, CA, USA), and *P* values smaller than 0.05 (*P* < 0.05) were considered a statistically significant difference. For the *post hoc* test of ANOVA, Tukey HSD was used as a default. For the analysis of some data that did not follow normal distribution, e.g., groups of predominantly monospermic eggs, two-tailed Mann-Whitney U Test was utilized (https://www.socscistatistics.com/), and the method used is indicated accordingly in the figure legend (referred to as U-test). 

## 3. Results

### 3.1. Nicotine Induces Polyspermy in a Dose-Dependent Manner 

For quantitative assessment of nicotine’s effect on polyspermy, sea urchin eggs were treated with increasing concentrations of nicotine (0−20 mM) prior to fertilization for 5 min. When the number of egg-incorporated sperm and the elevation of the fertilization envelope (FE) were examined 10 min after insemination (Figure 1), it was evident that both FE elevation and the egg-incorporated sperm counts were affected by nicotine pretreatment in a dose-dependent manner. With the increasing doses of nicotine, the frequency of the eggs displaying full-fledged elevation of FE at fertilization was progressively reduced, while the average number of egg-incorporated sperm was almost proportionally increased despite the seasonal batch-to-batch variability (Figure 1B). As FE elevation has been intuitively considered as a mechanism of mechanically blocking polyspermy in echinoderm [8], the detrimental effect of nicotine on FE elevation may be in part accountable for the increased rate of polyspermy. However, the relevance of the failed FE elevation to the observed increase in polyspermy was questionable in a couple of considerations. Firstly, at the nicotine doses 0.1 to 0.5 mM, the elevation of FE was substantially compromised (Figure 1B), but the number of the egg-incorporated sperm was virtually the same as the control (0 mM nicotine), which was mostly monospermic (Table 1). Secondly, the elevation of FE started to be completely inhibited at 2 mM of nicotine, but the number of egg-incorporated sperm continued to grow as the nicotine dose increased (Figure 1B). Hence, it appears that pretreatment of the eggs with nicotine induces supernumerary sperm entry by a dose-dependent mechanism, and that the integrity of the FE elevation is not the decisive factor determining the number of egg-incorporated sperm. 

### 3.2. Nicotine-Induced Polyspermy Is Neither Mimicked by Cholinergic Agonists Nor Inhibited by Antagonists of Nicotinic Acetylcholine Receptors 

For nicotine to effect polyspermy in sea urchin eggs, it requires a certain dose (Figure 1) and a considerable incubation time. Thus, *P. lividus* eggs exposed to relatively low doses of nicotine (e.g., 2–3 mM) were not always polyspermic unless they were preincubated for more than 5–10 min, which was comparable with the earlier reports using *Psammechinus miliaris* eggs [38]. The existence of this lag time implies that nicotine might either be converted to an active agonist or transmit its effect to some slow downstream effectors transducing the signal. The former possibility looked less likely in view of the fact that cotinine, the major metabolite of nicotine [39], did not induce polyspermy within the same concentration range (Table 1). We then tested whether nicotine causes polyspermy through its action on acetylcholine receptors (AChR), as sea urchin eggs (*L. variegatus*) are known to manifest characteristic ion flux activities of nicotinic AChR [32]. At variance with an earlier report based on the observation from another sea urchin species (*Psammechinus miliaris*) [40], however, we found that neither acetylcholine (ACh) nor its metabolism-resistant analogue carbachol reproduced the polyspermy-inducing effect of nicotine in the same experimental conditions as with nicotine. Virtually all *P. lividus* eggs pretreated with either of these two agonists of nAChR retained monospermy at fertilization within the same concentration ranges (Figure 2A). The inability of ACh to induce polyspermy was not because of the egg jelly coating that might have served as a diffusion barrier to preclude the access of the drug to its receptor in the plasma membrane. Indeed, in the eggs whose jelly coat was extensively removed [32], ACh again failed to induce polyspermy (Table 2). 

The failure of ACh to induce polyspermy was not likely to be because the nACh receptors on the egg surface are quickly desensitized by a large dose of ACh for a prolonged time in this experimental paradigm [41]. Firstly, ACh and carbachol also failed to induce polyspermy at a lower dose (10 µM ACh, 1.1 ± 0.5 sperm/egg; 10 µM carbachol, 1.0 ± 0.2 sperm/egg, n = 20). Secondly, if an excess amount of ACh or carbachol had desensitized nAChR while the polyspermy-inducing effect of nicotine was transmitted through the same receptor, pre-desensitization of the eggs with excess ACh or carbachol should be able to prevent nicotine from inducing polyspermy. However, that was not the case. Nicotine still induced polyspermy to the same extent (Figure 2B). In support of the idea that nAChR on the egg surface is irrelevant to the nicotine-induced polyspermy, antagonists of nAChR including the ones whose efficacy was demonstrated in sea urchin eggs [32] all failed to prevent nicotine from inducing polyspermy (Figure 2C). In the case of mecamylamine, the drug even marginally enhanced nicotine’s induction of polyspermy. Similarly, other classical competitive antagonists of nAChR at the cholinergic synapse (i.e., α-bungarotoxin, *d*-tubocurarine) had no inhibitory effect on the nicotine-induced polyspermy in *P. lividus* eggs at and above the range of doses displaying physiological efficacy (Appendix A). Hence, it is unlikely that nicotine induces polyspermy by activating nAChR on the surface of *P. lividus* eggs. 

### 3.3. Effects of Nicotine Pretreatment on Ca^2+^ Signaling in the Fertilized Eggs of P. lividus 

Sea urchin eggs pretreated with nicotine still displayed Ca^2+^ increases at fertilization, but certain aspects of the intracellular Ca^2+^ signaling were specifically altered (Figure 3). A control egg at fertilization normally responds to the fertilizing sperm with a cortical flash (CF) and a subsequent Ca^2+^ wave that propagates from the sperm interaction site to the antipode (Figure 3A). Whereas the peak amplitude of the Ca^2+^ wave was virtually unaffected by the nicotine pretreatment except for some marginal decrease at 100 µM, the amplitude of the CF in the eggs pretreated with nicotine was significantly reduced in a dose-dependent manner (Figure 3B, right panel). Although nearly 10 sperm may enter the egg pretreated with 20 mM nicotine (Figure 1B), the Ca^2+^ wave did not originate from as many sperm-entry sites (Figure 3C). The average number of the Ca^2+^ waves in these eggs was merely 1.27 ± 0.23 (n = 11), a value not significantly different from the control eggs with no nicotine treatment (1.04 ± 0.13, n = 77). Hence, the entry of multiple sperm is not accompanied by equally supernumerary Ca^2+^ waves in this species, suggesting that the Ca^2+^ stores and their release mechanism might have entered a refractory phase after the massive Ca^2+^ increase evoked by the first successful sperm. Interestingly, once arriving at the peak amplitude, the Ca^2+^ waves in the eggs pretreated with nicotine tended to decline to the basal level much faster (Figure 3A). After the peak (RFU_max_) was attained, it took 128.3 ± 66.2 sec (n = 66) for the Ca^2+^ wave to decline to ½ RFU_max_ in the eggs pretreated with 1 mM nicotine, which was much quicker than in the batch-matching control eggs (170.9 ± 95.0 s, n = 58, *P* < 0.01). This may be due, in part, to the nicotine-induced reorganization of Ca^2+^ stores including cortical granules and vesicles [11,42,43]. By contrast, the time lag between the onset of the CF and the initiation of the Ca^2+^ wave (defined as latent period, Figure 3C) was not affected by nicotine. Nonetheless, the speed of the wave propagation was somewhat affected, as judged by the length of the time required for the wave to traverse to the antipode. Evidently, in the eggs pretreated with 20 mM nicotine, the Ca^2+^ wave ran significantly faster (Figure 3C). Taken together, these results show that some physiological events taking place in the egg cortex following fertilization are markedly affected by nicotine pretreatment: CF (Figure 3B), Ca^2+^ wave propagation (Figure 3C), sperm entry (Figure 1A), and cortical granule exocytosis and the consequent elevation of FE (Figure 1B). Interestingly, this is the constellation of events that goes astray when the egg actin cytoskeleton is modified prior to fertilization [33,34,35]. 

### 3.4. Nicotine Hyperpolymerizes Subplasmalemmal Actin Filaments of the Sea Urchin Eggs in a Dose-Dependent Manner 

Since it is known that intracellular Ca^2+^ signaling and cortical granule exocytosis in the fertilized eggs of sea urchin and starfish are heavily influenced by the state of the cortical actin cytoskeleton [33,34,35,36,44,45], we tested whether the nicotine pretreatment of sea urchin eggs alters the structure of the cortical actin filaments. When F-actin in live eggs was visualized with microinjected Alexa-Phalloidin, it was evident that nicotine induced actin to hyperpolymerize, specifically in the subplasmalemmal zone (Figure 4). More importantly, the effect was dependent upon the nicotine dose and the time of incubation. Whereas the eggs treated with 2 mM nicotine displayed marked hyperpolymerization in the subplasmalemmal zone by 45 min (Figure 4A), the same extent of actin hyperpolymerization in the corresponding region was observed much earlier in the eggs exposed to higher doses of nicotine, i.e., within 10 min after the treatment with 5 mM nicotine (Figure 4B), and as early as 5 min after being exposed to 20 mM nicotine (Appendix A). As expected, nicotine-induced hyperpolymerization of subplasmalemmal actin was markedly inhibited by 3 μM latrunculin A (LAT-A), albeit not completely (Figure 4C). The dose-dependent hyperpolymerization of subplasmalemmal actin induced by nicotine was in line with the finding that the number of egg-incorporated sperm increased with the increasing dose of nicotine (Figure 1), raising the possibility that the nicotine-induced changes in the subplasmalemmal actin cytoskeleton may be the cause of the increased rate of polyspermy. To test this idea, prior to nicotine exposure, eggs were pretreated with specific agents that interfere with actin dynamics by binding to G-actin (Figure 5). Preincubation (15 min) of the eggs with low doses of LAT-A alone did not affect sperm entry, as most eggs fertilized afterwards were monospermic. However, the same pretreatment with LAT-A inhibited the polyspermy induced by 6 mM nicotine in a dose-dependent manner (Figure 5A). In corroboration of this finding, another actin drug Mycalolide B (MYC-B) had a similar inhibitory effect in a dose-dependent manner, albeit with less efficiency (Figure 5B). 

### 3.5. Nicotine Renders Egg surface Hyper-Receptive to Sperm 

A brief exposure of sea urchin eggs to high doses of nicotine markedly changed the topography of the egg surface, as monitored by scanning electron microscopy (Figure 6A). Of particular interest were microvilli, myriad dynamic microfilament-filled structures protruding from the egg surface, which are known to increase egg surface area and play important roles in sperm interaction [46,47]. However, microvilli density quantified on the randomly selected egg surface were not significantly increased after the nicotine treatment, as judged by the average number of microvilli present in the given areas of the visual field (80 μm^2^): 195.1 ± 8.6 (control eggs) versus 202.7 ± 11.2 (eggs treated with 5 mM nicotine for 5 min), n = 9, *P* = 0.1269. Hence, it was unlikely that the nicotine-induced polyspermy was caused simply because microvilli density increased. On the other hand, we noted that the entire surface of the nicotine-treated eggs was highly undulated presumably due to the reorganization of subplasmalemmal actin networks and the consequent dislocation of the cortical granules and vesicles. It is conceivable that the creation of numerous concaves on the egg surface may have enhanced the way microvilli interact with fertilizing sperm and thereby increased egg’s receptivity to the supernumerary sperm. Moreover, it appears that nicotine also changed the functionality of the cortical actin filaments. When viewed with confocal microscopy, 5 min preincubation with 20 mM nicotine visibly hyperpolymerized the subplasmalemmal actin filaments (Figure 6B, white arrow). When these eggs were fertilized in NSW, the actin filaments near the plasma membrane behaved in remarkably abnormal ways. In the control eggs, a few minutes after fertilization, the depolymerized actin in the subplasmalemmal zone began to repolymerize to form orderly actin filaments perpendicular to the plasma membrane and centripetally migrated toward the inner cytoplasm (Appendix A). In the nicotine-pretreated eggs, amid hyperpolymerized actin filaments in the subplasmalemmal zone, multiple fertilization cones extended strikingly thick actin bundles to incorporate supernumerary sperm, and there was no orderly centripetal migration of actin filaments (Appendix A). Therefore, nicotine not only changed the structure of the actin filaments at the egg surface, but also appeared to alter their functionality to render the egg hyper-receptive toward supernumerary sperm (see the Section 4, below).

### 3.6. Nicotine Has an Intracellular Target That Is Not Nicotinic AChR 

A neurotransmitter enclosed in synaptic vesicles, ACh is not expected to diffuse across the cellular membrane. On the other hand, because of its lipophilic nature, nicotine can pass through the plasma membrane even across the blood-brain barrier [48,49]. In theory, the target of nicotine in sea urchin eggs could be either intracellular nAChRs, which are assembled and transported through membrane trafficking [28], or those residing in a specialized subcellular locus, as was exemplified in neurons [50]. Then, the inability of ACh and carbachol to induce polyspermy (Figure 2) unlike nicotine (Figure 1) could be attributed to this fundamental difference in membrane permeation. To test the idea, ACh and carbachol were delivered into the cytoplasm by microinjection at a pharmacologically effective dose (10 μM, cytosolic concentration), but the treated eggs did not produce polyspermy at fertilization (Figure 7A). Conversely, antagonists of nAChR microinjected into the cytoplasm also failed to inhibit nicotine-induced polyspermy (Figure 7B). On the other hand, eggs microinjected with 20 mM of (−)nicotine manifested polyspermy at fertilization (Figure 7A), as with the eggs exposed to nicotine by bath incubation (Figure 1). Taken together, these findings suggest that nicotine may cause polyspermy by affecting its intracellular target, which is something other than nAChR. 

### 3.7. Nicotine Has a Direct Impact on Actin Dynamics 

The idea that the down-stream effector of nicotine actually resides inside the egg but is not nAChR is further supported by the finding that microinjected nicotine successfully induced the characteristic rearrangement of the subplasmalemmal actin cytoskeleton, while microinjected ACh and carbachol failed to do so (Figure 8). Furthermore, when administered by bath incubation, ACh and carbachol consistently failed to reproduce the nicotine-induced rearrangement of subplasmalemmal actin cytoskeleton in the sea urchin eggs (Figure 9A). Hence, while the mechanism by which nicotine promotes the hallmark rearrangement of the actin cytoskeleton in sea urchin eggs is yet to be known, nAChR is not likely to be the mediator, either on the egg surface or inside. Interestingly, nicotine exposure induced a small but distinct Ca^2+^ increase in the eggs with a considerable time lag, but again, ACh and carbachol failed to do so (Figure 9B). The Ca^2+^ increase appears to be due to the release from the intracellular stores and not from the external media, as judged by the fact that it is not affected by the removal of Ca^2+^ from the media (Figure 9B). This Ca^2+^ increase uniquely induced by nicotine is concomitant with the F-actin changes, raising the possibility that the Ca^2+^ increase is linked to the cytoskeletal change. Alternatively, nicotine might be able to induce direct rearrangement of the actin cytoskeleton independent of the Ca^2+^ increase. In view of the fact that polyamine often induces actin polymerization [51], it was conceivable that nicotine, essentially a diamine, might directly act on actin to enhance polymerization. To explore this idea, we tested whether nicotine directly affected polymerization kinetics of actin by utilizing an in vitro assay based on fluorimetry [52]. As shown in Figure 10, the quadruplicate samples of pyrene-conjugated G-actin readily underwent polymerization, charting a highly reproducible trajectory and reached the steady state phase in about 30 min (Figure 10A; note that the error bars are small enough to be masked by the green filled squares). In the presence of 24 µM nicotine, the actin monomers polymerized appreciably faster during the growth phase of polymerization (Figure 10A, brown open squares). Thus, until 27 min, the average of the fluorescence levels of the quadruplicate samples treated with nicotine was significantly higher than that of the samples without nicotine at each and every time point (*t*-test: *P* < 0.0001 to *P* < 0.05). After 28 min, the net polymerization status reached the steady state, and the fluorescence level remained the same with or without nicotine. This difference was not due to a contribution from the intrinsic fluorescence of nicotine, which was invariably at the background fluorescence levels (Figure 10, gray triangles). Hence, it was evident that the presence of nicotine significantly accelerated actin polymerization in vitro. On the other hand, nicotine also delayed depolymerization of F-actin. In the depolymerization assay, the polymerized F-actin gradually lost fluorescence as it gets depolymerized with time. The trajectories of the depolymerization process in the quadruplicate samples were also highly reproducible (Figure 10B, green filled squares). Interestingly, in the presence of 24 µM nicotine, the depolymerization process was considerably delayed (Figure 10B, brown closed squares). In the parallel experiment on the same plate, addition of 1.2 µM of cofilin (actin-depolymerizing protein) to the F-actin mixture strikingly accelerated the depolymerization process (Figure 10B, see the difference between green circles and green squares). When 24 µM nicotine was added to the F-actin mixture 1 min before cofilin, however, the cofilin-induced depolymerization process was significantly inhibited (Figure 10B, brown open circles). Nevertheless, if nicotine was added after or together with cofilin, the inhibitory effect of nicotine on cofilin was not observed. Taken together with other results showing that nicotine induces drastic structural changes of the actin cytoskeleton inside sea urchin eggs (Figure 4 and Figure 8), this observation in an isolated system of the purified proteins in test tubes raises an intriguing possibility that nicotine might directly target the actin cytoskeleton inside the cell. In theory, such a mechanism by which nicotine modifies the actin cytoskeleton either by enhancing its polymerization kinetics or by protecting the actin filaments from depolymerization could be entirely independent of the cholinergic pathway.

## 4. Discussion

Nicotine is a pyridine alkaloid that is enriched in tobacco leaves. As with many other naturally occurring alkaloids, nicotine may interact with its cognate molecules and exert its physiological impacts on the target animals in the ecological field. This is why the nicotine in tobacco leaves has been viewed as a chemical defense mechanism, and thus was used as an insecticide until recently [53,54]. Nicotine is well-known as the specific agonist of a class of AChR that mediates synaptic transmission in the neuromuscular junction and between neurons in mammalian brains [27,28,31]. While it has been suggested that nicotine may interact with the ACh-binding site of the receptor because nicotine’s conformation around the basic nitrogen and its electronic makeup are quite similar to those of ACh [48], it remains an open question as to whether or not AChR is the only physiological target of nicotine in the animal kingdom. 

Since soon after the first demonstration of fertilization as a process of sperm’s entry into the egg, it has been known that familiar alkaloids like nicotine and caffeine can induce polyspermy in sea urchin [23,24]. While the mechanism by which nicotine induces multiple sperm entry into the fertilized eggs of sea urchin is not yet fully understood, it has been predicted that nicotine treatment of the eggs either increases the rate of successful collision between the egg and sperm or inhibits the mechanism that prevents polyspermic fertilization [55]. As a mechanism for preventing polyspermic fertilization, several models have been suggested, including ones involving egg-released protease [56], egg actin cytoskeleton [36,57], FE elevation [8,12], and the changes of egg membrane’s electrical properties [8,15]. Notably, studies on voltage clamped sea urchin eggs have suggested that electrophysiological response, formation of fertilization cones (comprising F-actin), and the subsequent sperm entry are all sensitive to the membrane potential of the eggs, which is most permissive at −10 mV [58,59]. Thus, the idea that has gained momentum and that has been widely accepted is that the swift change of the membrane potential and the elevation of the FE serve as the fast and slow mechanisms for blocking polyspermy, respectively [8]. 

The primary goal of the present study was to understand how nicotine induces polyspermy in sea urchin eggs. Earlier findings that nicotine both lowers the amplitude of the fertilization potential and produces thinner FE were set forth as the reason that nicotine pretreatment of the eggs led to polyspermy [25]. We have confirmed that nicotine pretreatment causes egg surface changes that lead to the failure of FE elevation, but the actual number of the egg-incorporated sperm did not closely match the extent of failure in FE elevation. That is, the failure of FE did not necessarily lead to polyspermy in the eggs exposed to low doses of nicotine, and the egg-incorporated sperm count continued to increase when the nicotine dose was increased beyond the level above which FE elevation is completely blocked (Figure 1). This finding indicated that failure of FE elevation is not the primary cause of nicotine-induced polyspermy, and that there should be another layer of control that decides the number of sperm entry. This observation does not rule out the possibility that the mechanism fine-tuning the electrophysiological response of the eggs at fertilization is somehow impaired by nicotine in a way that allows multiple sperm entry. However, this model does not provide much explanation for the dose-dependent increase of egg-incorporated sperm, and is not supported by the results of a recent study showing that *P. lividus* eggs fertilized in the conditions lowering the fertilization potential are mostly monospermic [60,61]. Hence, our observation of dose-dependent effects of nicotine on polyspermy was not convincingly explained by the prevailing models based on the fast electric block and slow mechanical block to polyspermy.

As mentioned previously, actin drugs induce polyspermy and other anomalies in sperm entry [36,57], and it has been repeatedly demonstrated that starfish and sea urchin eggs, whose cortical F-actin is altered by various methods, tend to be polyspermic at fertilization [11,33,34,35,36,37,40,62]. Thus, we tested the idea that nicotine-induced polyspermy in sea urchin eggs may be linked to the actin cytoskeleton in the egg surface, and found that many aspects of nicotine-induced polyspermy in sea urchin eggs are mirrored by the cortical actin changes induced by nicotine. Firstly, as with the number of egg-incorporated sperm, which increased with the nicotine dose, the extent of nicotine-induced hyperpolymerization of actin filaments in the subplasmalemmal zone was also dose-dependent (Figure 4, Appendix A). That is, after a given time, such as 5 min incubation, eggs exposed to higher doses of nicotine exhibited more extensive actin polymerization and larger numbers of sperm incorporations at fertilization. Secondly, it appears that nicotine evokes a rather slow process that renders the egg polyspermic at fertilization. In this regard, it is important to note that nicotine-induced changes of the cortical actin cytoskeleton also required comparable time whose length was dependent upon the doses of nicotine and the sensitivity of confocal microscopy (Figure 4 and Appendix A). It is also noteworthy that nicotine hyperpolarizes the resting membrane potential with about a 10 min time lag in *P. lividus* eggs [26], and the shift of the current-voltage relationship induced by nicotine required circa 7 min in *Strongylocentrotus purpuratus* eggs [25]. In line with the fact that nicotine-induced polyspermy requires preincubation of the eggs for a comparable length of time [38], these findings suggest that nicotine-induced polyspermy depends upon some slowly progressing downstream effects, such as cortical reorganization of the actin cytoskeleton. Thirdly, the fertilization cones in the fertilized eggs (Appendix A) were evidently deregulated in the eggs pretreated with nicotine (Figure 6B and Appendix A). Finally, and most importantly, the nicotine-induced polyspermy was successfully inhibited by agents promoting actin depolymerization, namely LAT-A and MYC-B, in a dose-dependent manner (Figure 5). These data strongly suggest that nicotine-induced polyspermy is linked to the actin hyperpolymerization on the egg surface. In this context, as ion channel activities are often influenced by actin meshwork [63], it is also possible that the nicotine-induced actin cytoskeletal reorganization is upstream of the changes of electrical property of the egg membrane. 

In this communication, we mainly focused on the search for the downstream pathway that mediates the polyspermy-inducing effect of nicotine. To our surprise, several lines of evidence obtained in our study consistently suggested that this signaling pathway was not related to AChR, which is the well-known physiological target of nicotine. First, the *bona fide* agonists of the nicotinic AChR, i.e., ACh and its metabolically resistant analogue carbachol, failed to reproduce nicotine-induced polyspermy (Figure 2A). Arguing against the possibility that the latter results are attributable to fast or ultrafast desensitization of AChR [41], nicotine was still able to induce polyspermy in the eggs pretreated with an excess amount of ACh or carbachol (Figure 2B). Second, all the inhibitors of nicotinic AChR tested in our study, i.e., both competitive and non-competitive antagonists, failed to inhibit nicotine-induced polyspermy (Figure 2C, Appendix A). Third, the potential involvement of some intracellular receptors for ACh [50] was also ruled out. When a physiologically effective dose of ACh or carbachol was delivered into the cytoplasm by microinjection, such a treatment did not render the eggs polyspermic at fertilization (Figure 7A). Conversely, the three antagonists of nAChR microinjected into the cytoplasm did not inhibit the nicotine-induced polyspermy, either (Figure 7B). However, the direct target of nicotine that eventually renders the eggs polyspermic at fertilization appears to reside inside the egg. When (−)nicotine was delivered by microinjection, thereby bypassing AChR on the cell surface, the egg incorporated multiple sperm at fertilization (Figure 7A). Therefore, the data altogether support the idea that nicotine traverses the plasma membrane, unlike Ach, and acts on its direct target in the cytoplasm, which is not nicotinic AChR. This scenario is more than conceivable because, according to the Henderson–Hasselbalch equation, over 60% of nicotine in seawater (pH 8.2) is expected to exist in the membrane-permeant unprotonated form [49]. However, the actual concentration of nicotine in the cytoplasm of the eggs at the time of experiments is currently beyond our knowledge. Our conclusion that AChR is irrelevant to nicotine-induced polyspermy is at odds with an early report on another species of sea urchin, *Psammechinus miliaris* [40]. However, it should be noted that the experimental evidence that ACh induces polyspermy has never been formally presented in that work or elsewhere, to the best of our knowledge. Furthermore, the claim that nicotine-induced polyspermy is mitigated by an inhibitor of nAChR, *d*-tubocurarine (1.5 mg/mL, circa 2.4 mM), should be taken with caution because the drug at the same concentration apparently has toxic effects on the eggs of *P. lividus*. 

Given that ACh is irrelevant to polyspermy, the question remains on how nicotine induces polyspermy in a non-cholinergic pathway. Does nicotine interact with some proteins other than AChR? In the case of rat, nearly 0.03% of the brain’s total proteins showed some affinity to nicotine, but most of them have not been characterized, except for the subunits of nicotinic AChR [64]. Thus, to date, any physiological effect of nicotine transmitted through an AChR-independent pathway is hardly known. Nonetheless, in this study, we found some evidence that nicotine may directly or indirectly interfere with the polymerization and depolymerization kinetics of F-actin, while ACh and carbachol failed to do so. Indeed, the subplasmalemmal actin filaments underwent impressive hyperpolymerization in the eggs exposed to nicotine in a dose- and time-dependent manner (Figure 4), while the two agonists of AChR, i.e., ACh and carbachol, did not show such an effect, either by incubation (Figure 9A) or by microinjection (Figure 8). Thus, the unique effect of nicotine on F-actin polymerization seems to be independent of the nAChR that may exist on the egg surface [32] or in the intracellular domain. The rearrangement of the actin cytoskeleton by nicotine has been reported in a variety of cell types such as adrenal chromaffin cells [65,66,67], neutrophils [68], vascular endothelial cells [69,70], retinal pigment epithelial cells [71], osteoclasts [72], or even Amoeba [73], but its mechanism is mostly unclear or was attributed to the activation of nAChR. In the present study, we demonstrated that nicotine induces actin hyperpolymerization in the eggs and can directly accelerate the polymerization kinetics of purified actin in test tubes (Figure 10A). Furthermore, when added to the pre-assembled F-actin, nicotine not only significantly delays its depolymerization rate, but also protects them from the action of cofilin (Figure 10B), the classical actin-binding protein that promotes depolymerization of F-actin [74]. Although obtained with actin molecules purified from rabbit muscle, these data from in vitro studies are in good agreement with the observations made in the living eggs exposed to nicotine (Figure 4, Figure 8 and Figure 9). While it is an open question as to whether or not nicotine directly acts on actin and actin filaments in a living cell, it is conceivable that such a shift in actin dynamics taking place during the early the stage of actin nucleation might result in amplified consequences. To our knowledge, this is the first demonstration that nicotine by itself can directly enhance actin polymerization either by accelerating its polymerization kinetics, or by attenuating its depolymerization rate. The latter event can take place either by directly inhibiting actin disassembly or indirectly by interfering with the action of an actin-binding protein. In this regard, it would be of keen interest for a future study to test whether nicotine interferes with the action of other actin-binding proteins in vitro*,* and whether the nicotine’s effect on living cells can be affected by these proteins. Based on several lines of evidence, we propose that nicotine induces polyspermy by interfering with actin kinetics directly or indirectly, and that such cytoskeletal changes may facilitate engulfing of multiple sperm. Indeed, the more nicotine was added to the media, the more severe hyperpolymerization took place in the subplasmalemmal zone (Figure 4), and the more sperm entered the eggs at fertilization (Figure 1). The presence of extraordinarily dense actin meshwork underneath the plasma membrane is expected to facilitate the formation of supernumerary fertilization cones. Furthermore, after the nicotine treatment, the egg’s surface topography was impressively altered, creating numerous concaves. This contour change of the egg, which momentarily occurs during normal fertilization, might also make it easier for the egg to capture supernumerary sperm. In agreement with this conclusion, it bears emphasis that the interference with actin dynamics in sea urchin, starfish, and clam eggs all led to an increased rate of polyspermy [33,35,75]. 

Interestingly, we also noted that nicotine uniquely induced a distinct but delayed Ca^2+^ release from the internal store, which is independent of AChR or external Ca^2+^ in the media (Figure 9B). The temporal profile of this Ca^2+^ increase was in parallel with the changes of the cortical actin cytoskeleton (Figure 9), raising the possibility that nicotine may increase Ca^2+^ in some unknown mechanism, and the resulting Ca^2+^ increase caused the F-actin rearrangement in the egg cortex [76]. However, it is less likely that this Ca^2+^ signal contributes to the nicotine-induced polyspermy, because it takes place after the egg is rendered receptive toward supernumerary sperm. Alternatively, it cannot be ruled out that this Ca^2+^ increase might be due to the concomitant decrease of the actin filaments deep to the subplasmalemmal layers (Figure 9A), as depolymerization of Ca^2+^-containing actin filaments could directly release Ca^2+^ ions [57,63,77,78,79]. The nature of the nicotine-induced Ca^2+^ increase and its causal relationship with the concomitant changes of the actin cytoskeleton awaits further investigation in future studies. 

In conclusion, our study implies that the biological function of nicotine may have been narrowly interpreted and over-restricted to its relationship with AChR and the cholinergic pathway, while this familiar alkaloid can exert many other unknown effects in a novel pathway, as we demonstrated here. It is worth noting that nicotine induces both F-actin changes and polyspermy in starfish (*Astropecten aranciacus*) eggs (Appendix A), and perhaps in the eggs of other animal species. Although it has not been established whether nicotine causes polyspermy in humans, it has been reported that smoking is a significant risk factor for spontaneous abortion, which is due to chromosomal mismatch, as is caused by polyspermy [80,81,82,83]. Although the nicotine concentration in the blood of smokers is orders of magnitude lower than the conditions used in this study, its presence is rather chronic [84]. In this context, nicotine matters to the issues of human reproduction and other health concerns, as its effect could unfold in several different ways, including the novel pathway featured in this study. 

## Figures and Tables

**Figure 1 cells-09-00063-f001:**
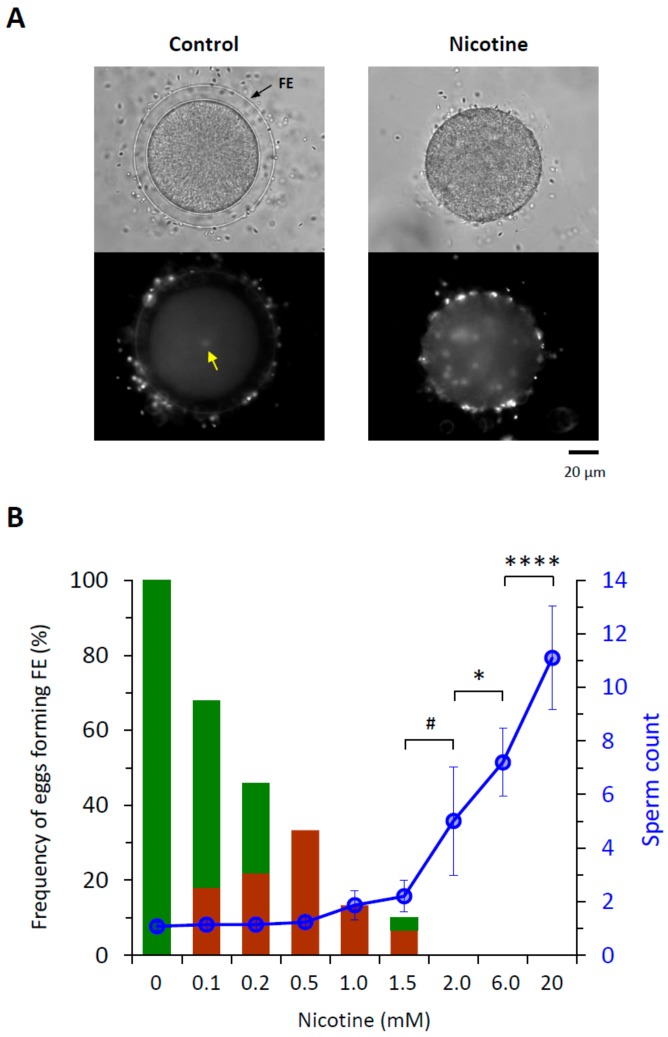
Nicotine induces polyspermy in a dose-dependent manner. *P. lividus* eggs were incubated for 5 min in the presence of various concentration of nicotine. About 10 min after fertilization with Hoechst 33342-prestained sperm, the zygotes were examined with a CCD camera to monitor sperm entry and the elevation of the fertilization envelope (FE). (**A**) Bright field view and the epifluorescence photomicrographs (bottom) showing the control egg and the egg exposed to 6 mM nicotine prior to fertilization. Whereas a single sperm entered the control egg (yellow arrow), numerous sperm were incorporated into the nicotine-exposed eggs. (**B**) Quantification of the egg-incorporated sperm and the extent of FE elevation. Green bars in the histogram represent the eggs with full-fledged elevation of FE, while brown bars stand for the eggs showing thin modest elevation of the FE. Error bars indicate standard deviation of the sperm counts averaged from multiple eggs specified in Table 1. U-test: ^#^
*P* < 0.05, * *P* < 0.01, **** *P* < 0.00001.

**Figure 2 cells-09-00063-f002:**
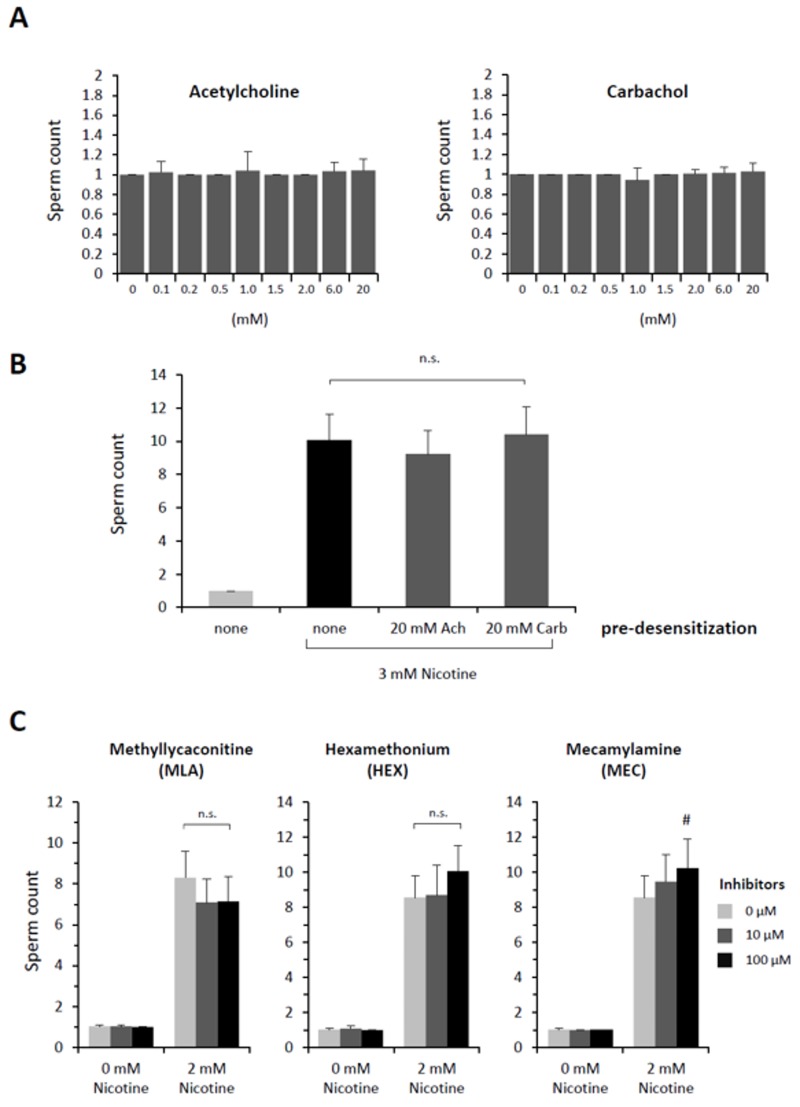
Acetylcholine receptors are not involved in nicotine-induced polyspermy in *P. lividus*. (**A**) Prior to fertilization, eggs were pretreated with either acetylcholine (ACh) or carbachol at the same range of concentration (80 to 150 eggs for a given concentration) and for the same incubation time (5 min) as in the nicotine experiments in Figure 1. The number of Hoechst 33342-prestained sperm incorporated into the egg was counted. (**B**) *P. lividus* eggs (n = 40 for each condition) were desensitized by excess amount of acetylcholine (ACh) or carbachol (Carb) for 5 min prior to exposure to 3 mM (−) nicotine (5 min). The egg-incorporated sperm were counted 10 min after fertilization. One-way ANOVA: n.s. (differences statistically non-significant). (**C**) Eggs were pretreated with various inhibitors of ACh receptors 5 min before the addition of nicotine (2 mM) or NSW. Five minutes later, eggs were fertilized (n = 40 for each condition), and the number of egg-incorporated sperm were counted. ^#^ Significantly different from the eggs pretreated with 2 mM nicotine without the inhibitor: Tukey HSD test, ^#^
*P* < 0.05.

**Figure 3 cells-09-00063-f003:**
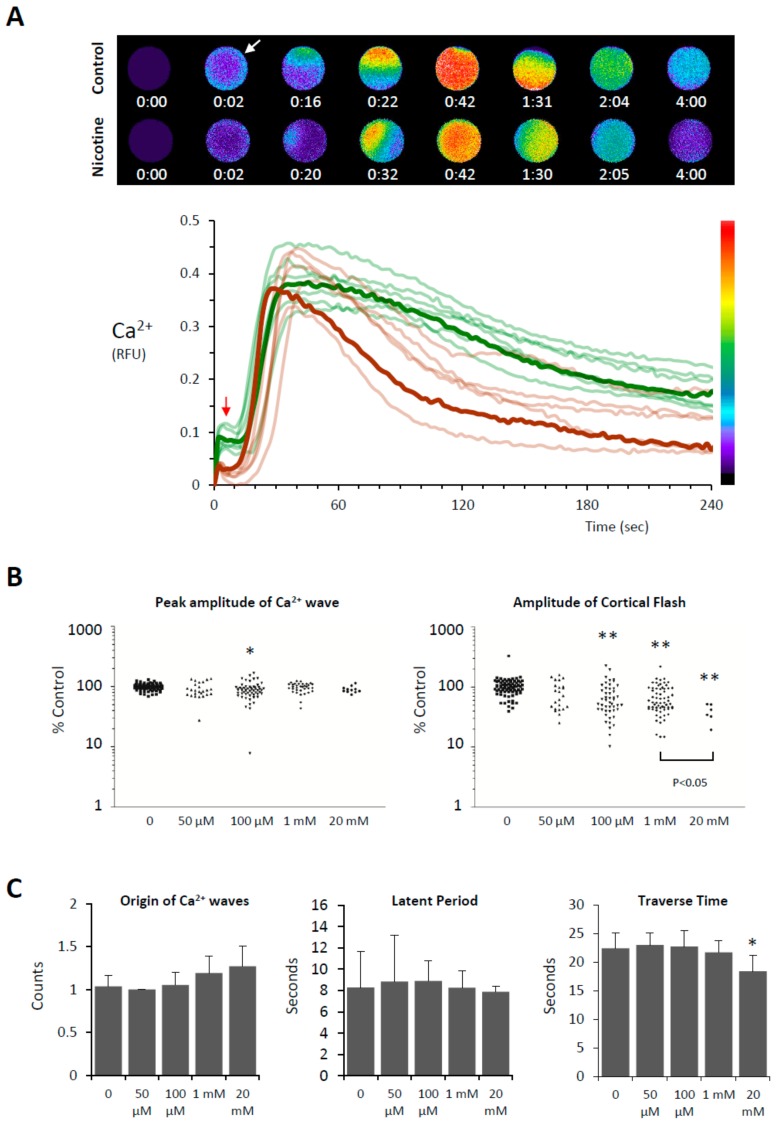
Effects of nicotine pretreatment on the Ca^2+^ signaling in the fertilized eggs of *P. lividus*. The Ca^2+^ responses in the fertilized eggs pretreated with various concentrations of nicotine (5 min) were pooled from multiple experiments comprising 5 to 8 eggs for each nicotine dose. (**A**) The pseudo-colored relative fluorescence images at the top panel represent the sites of instantaneous Ca^2+^ increases at the key time points in the eggs fertilized with or without nicotine pretreatment (20 mM). The graph shows the trajectory of the intracellular Ca^2+^ levels based on relative fluorescence unit (RFU) defined in Materials and Methods. Arrows indicate CF (cortical flash). (**B**) Effects of nicotine on the CF and the peak of the Ca^2+^ wave. The peak amplitudes of CF and the Ca^2+^ wave in individual eggs were normalized in reference to the corresponding average values in the control eggs (no nicotine pretreatment) from the same batch of experiment. (**C**) Effects of nicotine on other aspects of the Ca^2+^ response at fertilization: counts of Ca^2+^ wave origins, latent period (i.e., time interval between the CF and the initiation of the Ca^2+^ wave), and the time required for the Ca^2+^ wave to travel from the sperm interaction site to the antipode (traverse time). Values significantly different from the control (0 mM nicotine treatment, Tukey HSD): ** *P* < 0.001, * *P* < 0.01.

**Figure 4 cells-09-00063-f004:**
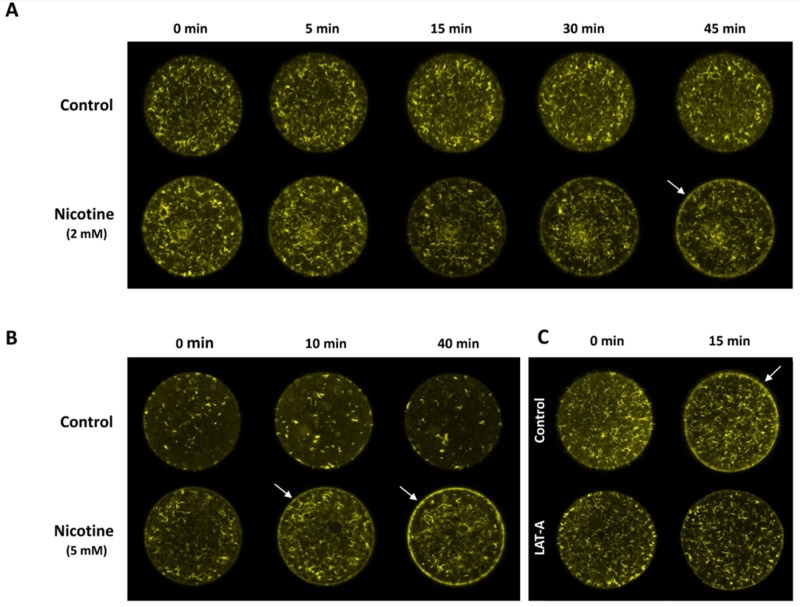
Effects of nicotine on the structure of the actin cytoskeleton in the subplasmalemmal region of sea urchin eggs. *P. lividus* eggs microinjected with AlexaFluor 568–phalloidin (10 µM, pipette concentration) were incubated with different doses of nicotine, and the changes of F-actin were monitored on the equatorial plane of the live individual egg by confocal laser scanning microscope. (**A**) 2 mM nicotine, (**B**) 5 mM nicotine. The eggs from the same batch without nicotine exposure were used as the control. For each experiment, the moment immediately before nicotine or seawater addition was taken as t = 0. (**C**) Latrunculin A (LAT-A) attenuates nicotine-induced rearrangement of cortical F-actin. *P. lividus* eggs microinjected with AlexaFluor 568–phalloidin were incubated in the presence of 3 μM LAT-A or 0.1% DMSO (control) for 20 min. After rinsing in NSW, the eggs were exposed to 10 mM (−)nicotine, and the changes of F-actin were monitored. Each image represents similar results obtained from 4–7 eggs in the given condition.

**Figure 5 cells-09-00063-f005:**
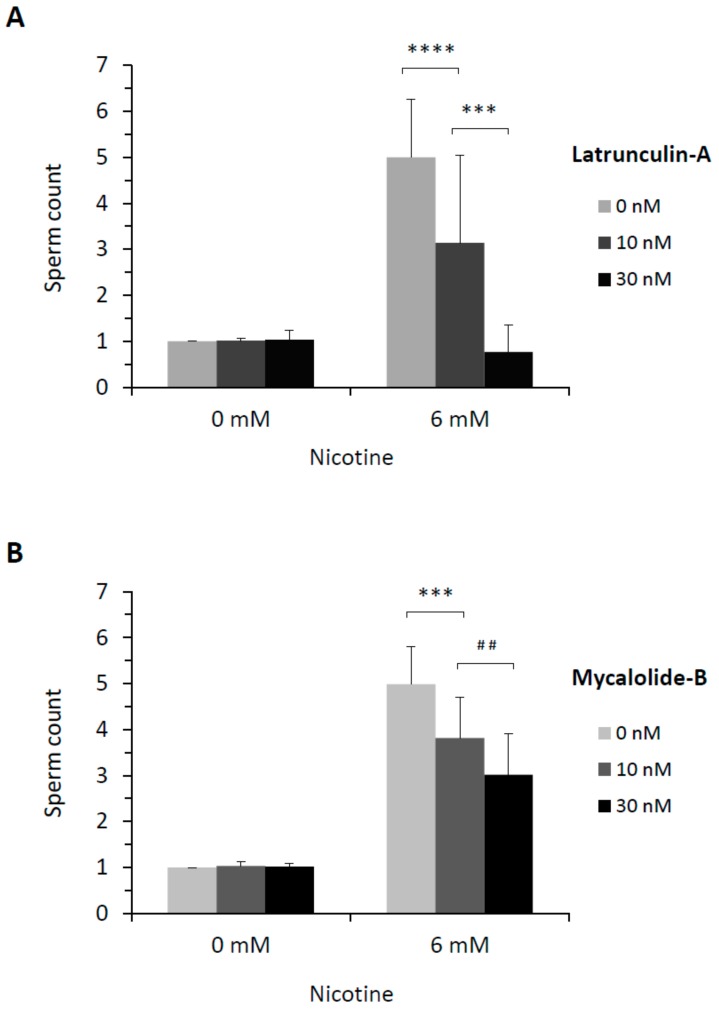
Nicotine-induced polyspermy is inhibited by actin drugs in a dose-dependent manner. *P. lividus* eggs were pretreated with latrunculin A (**A**) or mycalolide B (**B**) for 5 min and rinsed twice in NSW prior to the exposure to 6 mM nicotine (5 min). The eggs were then fertilized with Hoechst 33342-prestained sperm, and the number of egg-incorporated sperm was counted 10 min after fertilization. Control eggs without exposure to actin drugs (0 nM) received the same pretreatment with the vehicle of the drugs (0.1% DMSO). Data were pooled from 4 (latrunculin-A) or 3 (mycalolide B) independent experiments. Each treatment in one experiment comprised 20 eggs from the same batch (n = 80 or 60, respectively). U-test: **** *P* < 0.00001, *** *P* < 0.0001, ^##^
*P* < 0.025.

**Figure 6 cells-09-00063-f006:**
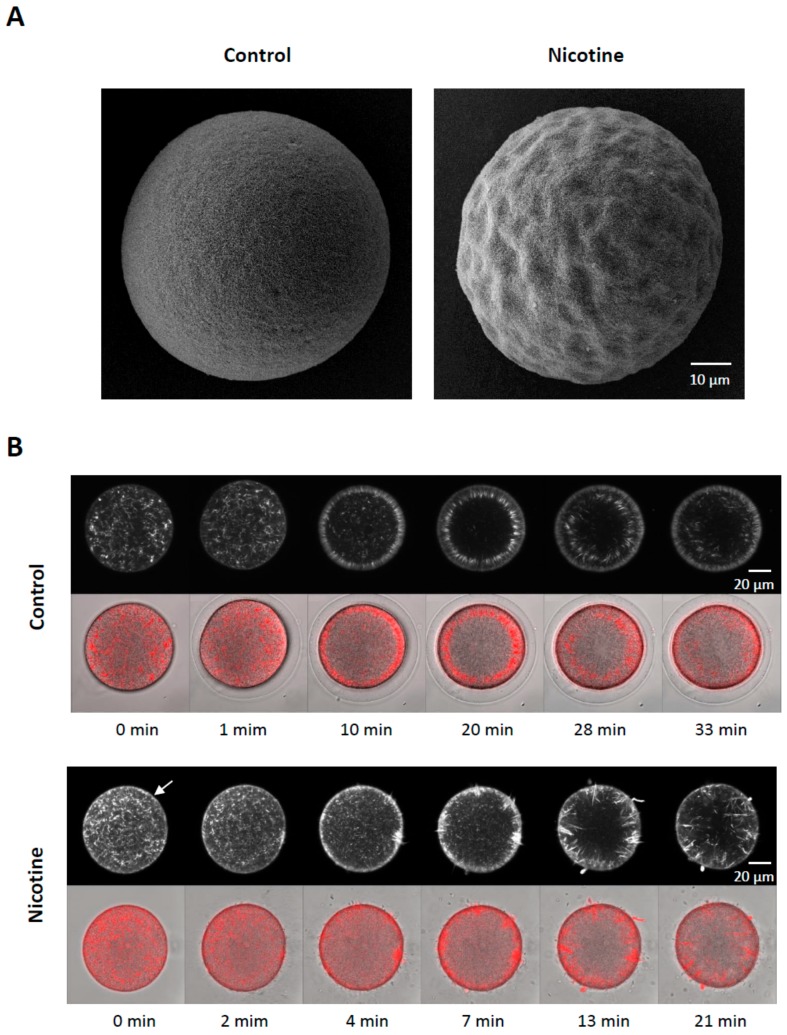
Nicotine renders egg surface hyper-receptive to supernumerary sperm. (**A**) Comparison of the cell surface topography of the jelly free eggs by scanning electron microscopy after pretreatment of the eggs in the presence or absence of 5 mM nicotine (5 min). (**B**) Effects of nicotine on the mobilization of cortical actin filaments following fertilization. *P. lividus* eggs were microinjected with AlexaFluor568-Phalloidin and subsequently incubated in the presence or absence of 20 mM nicotine for 5 min. The eggs were then fertilized in fresh NSW and the distribution of actin filaments were continuously monitored by confocal microscopy. Merged images with the bright field view show lack of fertilization envelope elevation in nicotine-pretreated eggs. The post-fertilization time points were shown below the merged view of the same egg at the given moment. The images labeled 0 min refer to the ones captured immediately before the addition of sperm.

**Figure 7 cells-09-00063-f007:**
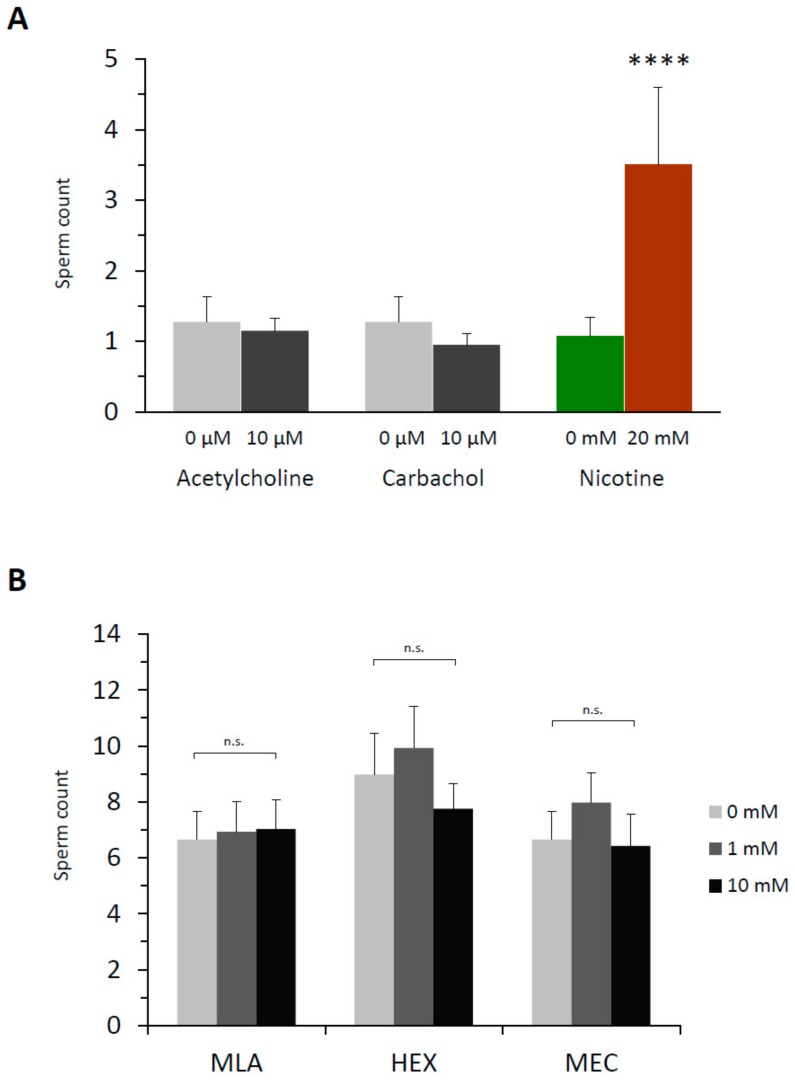
Nicotine has an intracellular target that is not a nicotinic AChR. (**A**) *P. lividus* eggs (n = 40 for each condition) were microinjected with 1 mM (pipette concentration) of ACh or carbachol. Control eggs (0 mM) were microinjected with distilled water. For comparison, eggs were microinjected with 20 mM (−)nicotine (cytosolic concentration). After 5 min incubation, the eggs were fertilized by Hoechst 33342-prestained sperm. The number of egg-incorporated sperm was counted 10 min after insemination. U-test: **** *P* < 0.00001. (**B**) *P. lividus* eggs were microinjected with inhibitors of nicotinic AChR at various doses, i.e., 10 μM or 100 μM (cytosolic concentration): methyllycaconitine (MLA), hexamethonium (HEX) or mecamylamine (MEC). Control eggs (0 μM) were microinjected with distilled water. After 5 min incubation, the eggs were exposed to 6 mM nicotine for 5 min and fertilized. Egg-incorporated sperm were counted 10 min after insemination. In both panels, data were pooled from two independent experiments comprising 20 eggs for each treatment condition. One-way ANOVA test: n.s., non-significant.

**Figure 8 cells-09-00063-f008:**
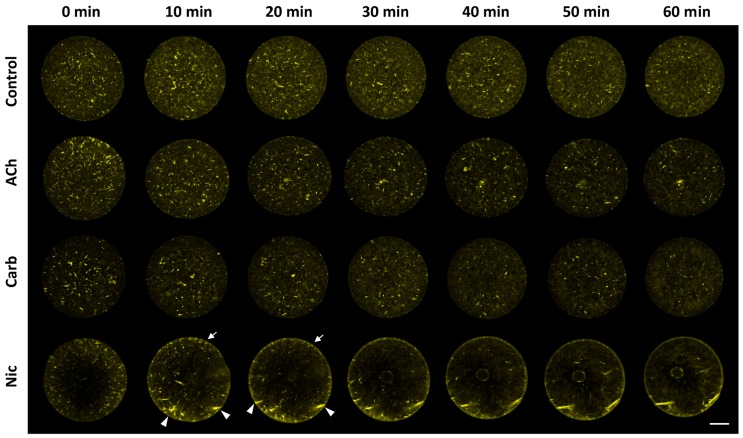
Effects of microinjected ACh, carbachol and nicotine on cortical F-actin of sea urchin eggs. *P. lividus* eggs were microinjected with Alexa-Phalloidin (10 μM, pipette concentration). After 15 min, the eggs were microinjected again with ACh or carbachol (1 mM, pipette concentration), and the changes of the actin filament in the same living eggs were monitored by confocal microscopy at 10 min intervals. The confocal images immediately before the second microinjection were considered as t = 0 min. For comparison, the second microinjection of some eggs was made with 1 mM (pipette concentration) of (−)nicotine, or with distilled water (negative control). Please note that only in the eggs microinjected with nicotine, the subplasmalemmal F-actin evidently started to get hyperpolymerized by 10–20 min (arrows) or to form thick F-actin bundles (arrowheads). Scale bar = 20 μm. Each image represents similar results obtained from 4–5 eggs in the given condition.

**Figure 9 cells-09-00063-f009:**
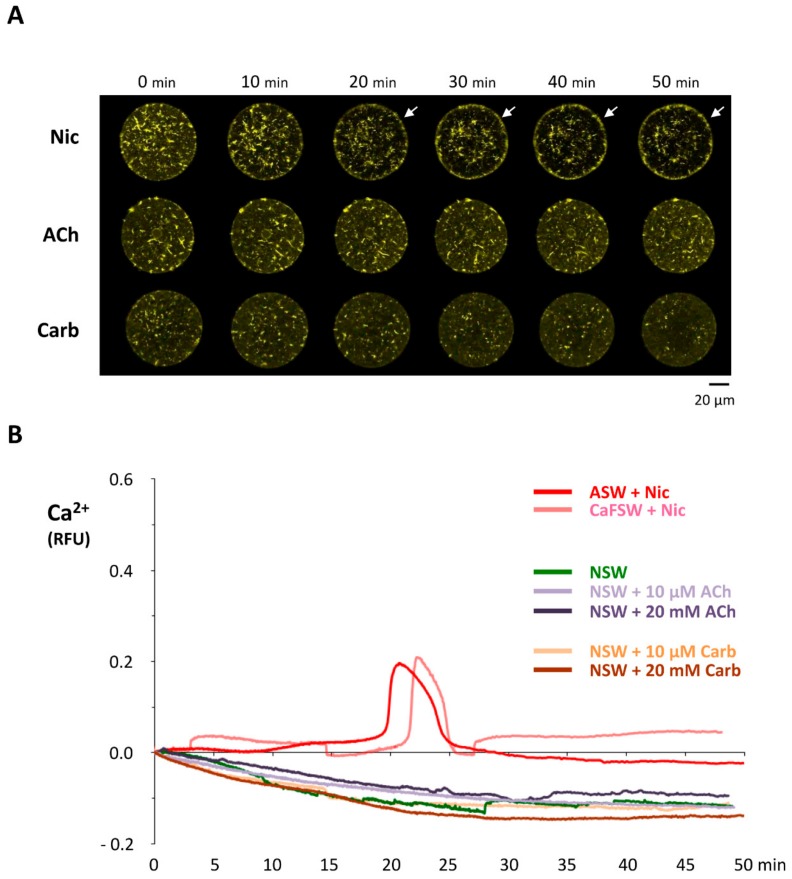
Unique effects of nicotine on subplasmalemmal actin filaments and the intracellular Ca^2+^ levels. (**A**) *P. lividus* eggs were microinjected with 10 μM Alexa-Phalloidin. After 15 min incubation, the eggs were exposed to 20 mM nicotine (Nic), acetylcholine (ACh), or carbachol (Carb), and the changes of the actin filaments in the same living eggs were monitored by confocal microscopy at 10 min intervals. Note the hyperpolymerization of subplasmalemmal F-actin in the nicotine-incubated eggs that was much pronounced by 20 min (arrows). Each image represents similar results obtained from 8–10 eggs in the given condition. (**B**) A delayed increase of intracellular Ca^2+^ in the eggs exposed to 20 mM Nicotine in the 10 mM Ca^2+^ artificial seawater (ASW), and Ca^2+^-free seawater (CaFSW). *P. lividus* eggs microinjected with Calcium Green were exposed to nicotine and other agonists of nAChR. Unlike nicotine, small and large doses of ACh and carbachol commonly fail to induce the Ca^2+^ changes.

**Figure 10 cells-09-00063-f010:**
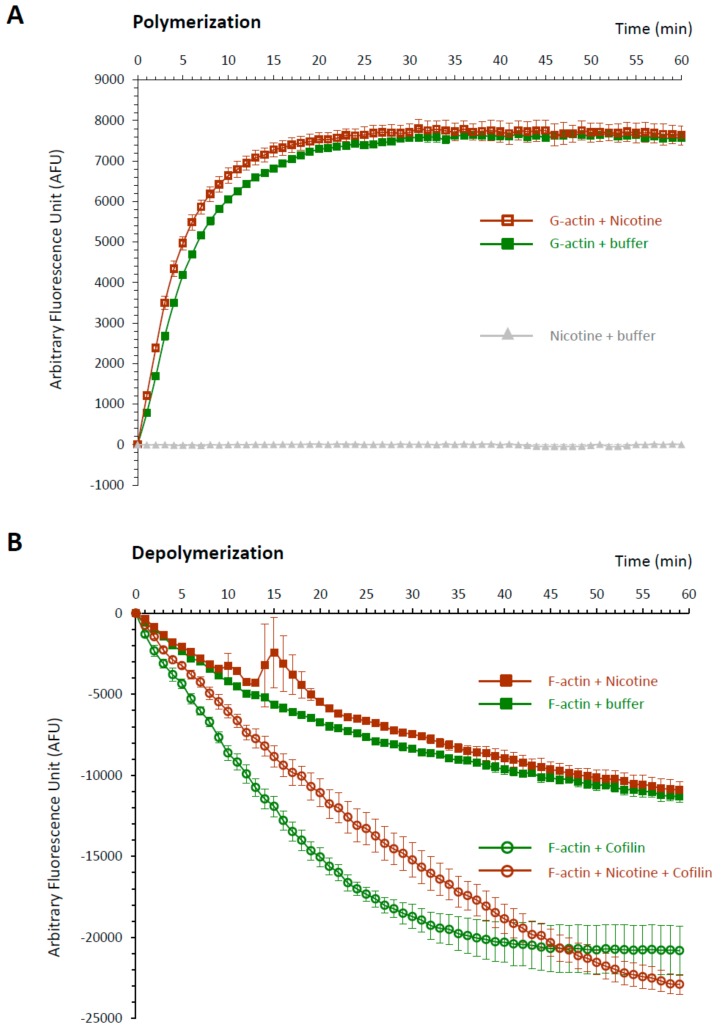
Direct impact of nicotine on the polymerization and depolymerization dynamics of actin. (**A**) Actin polymerization kinetics was assessed following the procedures described in Materials and Methods. The trajectories of the actin polymerization kinetics in the presence or absence of 24 μM nicotine were marked by open brown squares and filled green squares, respectively. Each data point in every curve represents mean ± SD obtained from quadruple samples. Gray triangles represent the virtually unchanging background levels of fluorescence in the wells containing the buffer and nicotine (note that the error bars are small enough to be masked by the symbols). (**B**) The depolymerization kinetics of pre-polymerized F-actin in the presence (brown squares) or absence (green squares) of 24 μM nicotine. Please note that addition of 1.2 μM cofilin to the reaction mixture steeply accelerated the depolymerization kinetics (open green circles). When 24 μM nicotine was introduced to the F-actin mixture 1 min prior to the cofilin addition, cofilin was appreciably less effective in depolymerizing F-actin (brown open circles).

**Table 1 cells-09-00063-t001:** Effects of nicotine and cotinine on the fertilization of *P. lividus* eggs.

**Nicotine (mM)**	**0**	**0.1**	**0.2**	**0.5**	**1**	**1.5**	**2**	**6**	**20**
**Egg-incorporated Sperm Count**	1.11 ± 0.33	1.14 ± 0.40	1.14 ± 0.40	1.23 ± 0.50	1.87 ± 1.07	2.2 ± 1.2	5.0 ± 4.0 *	7.2 ± 2.5 *	11.1 ± 3.9 *
**Eggs with Full/Partial FE elevation**	100%/0%	50%/18%	24%/22%	0%/33%	0%/13%	3.3%/6.7%	0%/0%	0%/0%	0%/0%
**n**	160	50	50	30	30	30	80	40	40
**Cotinine (mM)**	**0**	**0.1**	**0.2**	**0.5**	**1**	**1.5**	**2**	**6**	**20**
**Egg-incorporated Sperm Count**	1.0 ± 0.0	1.0 ± 0.0	1.05 ± 0.22	1.1 ± 0.31	1.0 ± 0.0	1.05 ± 0.22	1.0 ± 0.0	1.1 ± 0.31	0.55 ± 0.60 ^#^
**Eggs with Full/Partial FE elevation**	100%/0%	75%/25%	65%/35%	25%/45%	5%/75%	10%/20%	0%/0%	0%/0%	0%/0%
**n**	20	20	20	20	20	20	20	20	20

* Significantly different from the values in the control (no drug). U-test: * *P* < 0.00001, ^#^
*P* < 0.05.

**Table 2 cells-09-00063-t002:** Acetylcholine fails to induce polyspermy also in the dejellied eggs.

Acetylcholine (mM)	0	0.01	0.1	0.2	0.5	1	2	6	10	20
**Egg-incorporated sperm Count**	1.10 ± 0.31	1.10 ± 0.31	1.05 ± 0.22	1.05 ± 0.22	1.00 ± 0.0	1.05 ± 0.22	1.00 ± 0.0	1.00 ± 0.0	1.05 ± 0.22	1.00 ± 0.0
**n**	20	20	20	20	20	20	20	20	20	20

Note: As previously described by Ivonnet and Chambers (1997), *P. lividus* eggs were dejellied by repeated shaking in test tubes prior to 5 min exposure to various concentration of Acetylcholine. The eggs were fertilized with Hoechst 33342-prestained sperm, and the egg-internalized sperm were counted 10 min after fertilization.

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
