# Peer review of "Nicotine Induces Polyspermy in Sea Urchin Eggs through a Non-Cholinergic Pathway Modulating Actin Dynamics"

_cells, 2019, doi:10.3390/cells9010063_

Round 1

Reviewer 1 Report

See attached file

Author Response

The authors' point-by-point response to the Reviewer's comments was attached as a PDF file.  

Reviewer 2 Report

In this paper, authors analyze molecular events occurring in sea urchin eggs exposed to nicotine that induces polyspermy. They confirm literature data on the nicotine ability to induce polyspermy in sea urchin eggs. Interestingly, authors demonstrate that nicotine seems to be responsible for a rearrangement of cortical eggs cytoskeleton that makes eggs predisposed to the entry of more male gametes and that this takes place in a dose dependent manner. The experimental data here reported suggest that the effects of nicotine on sperm incorporation are not due to the bond with nicotinic receptors located on the eggs surface. Finally, in vitro tests about actin polymerization further suggest that polyspermy depends not only from membrane depolarization, but also from cytoskeletal rearrangement.

The paper is interesting and contribute to the open question about the block of polyspermy, but we cannot for now generalize author conclusions, we can say that nicotine induces polyspermy as a consequence of cytoskeletal modifications.

I suggest refining the language somewhere to avoid repetition of known concepts and to facilitate reading

I recommend acceptance after minor changes.

Author Response

(The authors gave the same response as above.)
